# Multiple valence bands convergence and strong phonon scattering lead to high thermoelectric performance in p-type PbSe

Yingcai Zhu [1], Dongyang Wang[1], Tao Hong[1], Lei Hu [2], Toshiaki Ina[3], Shaoping Zhan[1], Bingchao Qin[1], Haonan Shi[1], Lizhong Su[1], Xiang Gao[4] & Li-Dong Zhao [1,5✉]

Thermoelectric generators enable the conversion of waste heat to electricity, which is an effective way to alleviate the global energy crisis. However, the inefficiency of thermoelectric materials is the main obstacle for realizing their widespread applications and thus developing materials with high thermoelectric performance is urgent. Here we show that multiple valence bands and strong phonon scattering can be realized simultaneously in p-type PbSe through the incorporation of $AgInSe_2$. The multiple valleys enable large weighted mobility, indicating enhanced electrical properties. Abundant nano-scale precipitates and dislocations result in strong phonon scattering and thus ultralow lattice thermal conductivity. Consequently, we achieve an exceptional $ZT$ of ~ 1.9 at 873 K in p-type PbSe. This work demonstrates that a combination of band manipulation and microstructure engineering can be realized by tuning the composition, which is expected to be a general strategy for improving the thermoelectric performance in bulk materials.

[1] School of Materials Science and Engineering, Beihang University, Beijing 100191, China. [2] State Key Laboratory for Mechanical Behavior of Materials, Xi'an Jiaotong University, Xi'an 710049, China. [3] Research and Utilization Division, Japan Synchrotron Radiation Research Institute (JASRI/SPring-8), Sayo, Hyogo, Japan. [4] Center for High Pressure Science and Technology Advanced Research (HPSTAR), Beijing 100094, China. [5] Key Laboratory of Intelligent Sensing Materials and Chip Integration Technology of Zhejiang Province, Hangzhou 310051, China. ✉email: zhaolidong@buaa.edu.cn

The depletion of fossil fuels and the deteriorating environ-ment motivate the human beings to find sustainable and clean energy solutions. Thermoelectric devices can be used in energy harvesting from waste heat or be utilized in refrigera-tion, which is favorable for raising energy efficiency, attracting widespread attention from around the world. The efficiency of thermoelectric devices is largely determined by the figure of merit $ZT$ of their constituent thermoelectric materials, $ZT = \frac{S^2 \sigma T}{\kappa_e + \kappa_L}$, where $S$ represents the Seebeck coefficient, $\sigma$ is the electrical con-ductivity, $\kappa_e$ is the electronic contribution to the thermal con-ductivity, $\kappa_L$ is the lattice thermal conductivity, and $T$ is the absolute temperature, respectively. However, decoupling the interdependence between electrical and thermal transport prop-erties is a crucial but challenging issue for improving the ther-moelectric performance of materials. To achieve good electrical properties, various strategies such as band convergence[1–4], band sharpening[5], band alignment[6], carrier mobility optimization[7] and resonant states introduction[8] were adopted. On the other hand, materials with disordered or complex crystal structure[9,10], giant anharmonicity[11,12], metavalent bonding[13], and lone pair electrons[14] often exhibit intrinsic low lattice thermal conductivity, which are promising candidates for thermoelectric applications. Moreover, the lattice thermal conductivity can be largely sup-pressed by microstructural engineering, including nanoscale precipitates[15,16], dislocations[17,18], grain boundaries[19], and all-scale hierarchical architectures[20–22]. Therefore, a synergistic combination of electronic band modulation and microstructural engineering is expected to achieve advanced thermoelectric materials.

PbTe has long been used for mid-temperature power genera-tion, whereas the scarcity of element Te makes it expensive for wide applications. PbSe is a perfect substitute for expensive PbTe due to the earth-abundant element Se. Hitherto, only limited studies show that the $ZT$ of PbSe could reach 1.7[23–25], motivating us to search strategies to improve the thermoelectric properties of PbSe. The weighted mobility ($\mu_W = \mu(m^*/m_e)^{3/2}$) is a good descriptor for the inherent electrical performance of materials[26]. Multiple degenerate electronic bands enable large density-of-states effective mass $m^*$ without obvious effect on the carrier mobility ($\mu$)[1], facilitating the improvement of $\mu_W$. Indeed, the interplay of multiple bands enable large power factor or $\mu_W$ and thus ultrahigh $ZT$[27,28]. However, the two-band convergence is much difficult to realize due to the large energy offset between the valence band maximum (L) and the secondary valence band maximum ($\Sigma$) in PbSe and to date only limited works can pro-mote band convergence in it[24,29,30]. It is more challenging to achieve multiple bands convergence in PbSe.

The lattice thermal conductivity is another important para-meter for the thermoelectric performance indicated by the quality factor $B$ ($B \sim \mu_W/\kappa_L$)[31,32]. The introduction of materials with low lattice thermal conductivity in MTe (M = Pb, Ge) matrixes was proved to be an effective method to manipulate their thermal transport properties[33,34]. For example, the appearance of nano-dots in AgPb$_m$SbTe$_{2+m}$ (LAST) system is considered as the origin of their low lattice thermal conductivity and thus the enhanced thermoelectric performance[33]. Interestingly, the electrical pro-perties of materials can also be optimized in a similar way, such as in PbTe-AgInTe$_2$ (LIST)[35] and SnTe-AgInTe$_2$[36]. These enhanced performances motivate us to search strategies for optimizing the $\mu_W$ and $\kappa_L$ simultaneously.

In this work, beyond the two-band convergence between the L and $\Sigma$ bands, we found that a third valence band $\Lambda$ with a degeneracy $N_v = 8$ could be activated (Fig. 1a) through the incorporation of AgInSe$_2$ in the PbSe matrix doped with 2% Na (LISS). This three-band convergence tendency enables large

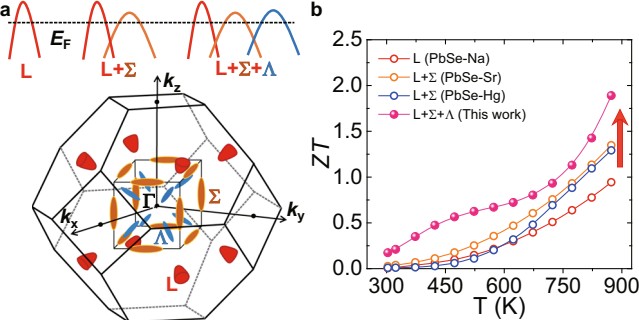

**Fig. 1 Multiple valence bands enable high ZT values in p-type PbSe.**
**a** Schmatic diagram of multi-bands (L, $\Sigma$, $\Lambda$) involvement in transport. The Brillouin zone shows that the degeneracies at the L, $\Sigma$, and $\Lambda$ points are 4, 12, and 8, respectively. **b** The activated third band $\Lambda$ enables higher $ZT$ values compared with the single-band and two-band PbSe-based materials.

weighted mobility. Additionally, local structure analysis by the x-ray absorption fine structure (XAFS) spectra indicates that about 80% of Ag and In atoms form AgInSe$_2$ in the system. Interestingly, AgInSe$_2$ is also a good thermoelectric material with intrinsic ultralow lattice thermal conductivity[37–39]. Nanoscale AgInSe$_2$ precipitates are revealed by the transmission electron microscopy (TEM), causing strong phonon scattering and hence resulting in ultralow lattice thermal conductivity. Therefore, a synergistic optimization of $\mu_W$ and $\kappa_L$ is realized. As a con-sequence, an exceptional high $ZT \sim 1.9$ is achieved at 873 K, which is much better than the single-band and two-band activated p-type PbSe-based materials[29,30] (Fig. 1b).

## Results

**Crystal structure**. The LISS compounds crystallize in cubic structure (Space group, *Fm-3m*), which is reflected by the x-ray diffraction (XRD) measurements that the XRD patterns can be indexed on the basis of cubic PbSe and no secondary phase is observed within the instrumental detection limit (Fig. 2a, b). The diffraction peaks tend to shift to higher angles with increment of AgInSe$_2$. Therefore, the lattice parameter ($a$) slightly decreases with increasing AgInSe$_2$ content (Fig. 2c), which may be attrib-uted to the smaller atomic radius of Ag, and In compared with that of Pb. This phenomenon also demonstrates that the AgInSe$_2$ is incorporated in the Pb$_{0.98}$Na$_{0.02}$Se matrix.

**Electrical transport properties**. The continuous decrease of the electrical conductivity with increasing temperature indicates a degenerate semiconducting property for LISS samples (Fig. 3a). Additionally, the electrical conductivity is suppressed significantly after the introduction of AgInSe$_2$. The electrical conductivity of Pb$_{0.98}$Na$_{0.02}$Se is as large as 3848 S/cm at room temperature, which declines to 774 S/cm for Pb$_{0.98}$Na$_{0.02}$Se−2.15% AgInSe$_2$ sample. To uncover this behavior, room temperature Hall mea-surements were performed. Obviously, the carrier concentration is reduced largely with increasing AgInSe$_2$ (Supplementary Fig. 1a), explaining the depressed electrical conductivity. The reduction of carrier concentration may be due to the formation of In$_{Pb}$ defects. These In$_{Pb}$ defects are shallow donors in PbSe[40], which will counteract with holes.

The Seebeck coefficient increases with elevated temperature for all samples and no saturate peak appears (Fig. 3b), demonstrating that no obvious bipolar effect occurs at high temperatures. The Seebeck coefficient is largely enhanced over the whole temperature range with the increment of AgInSe$_2$. Typically, the Seebeck coefficient of Pb$_{0.98}$Na$_{0.02}$Se is only 19.2 µV/K at room temperature,

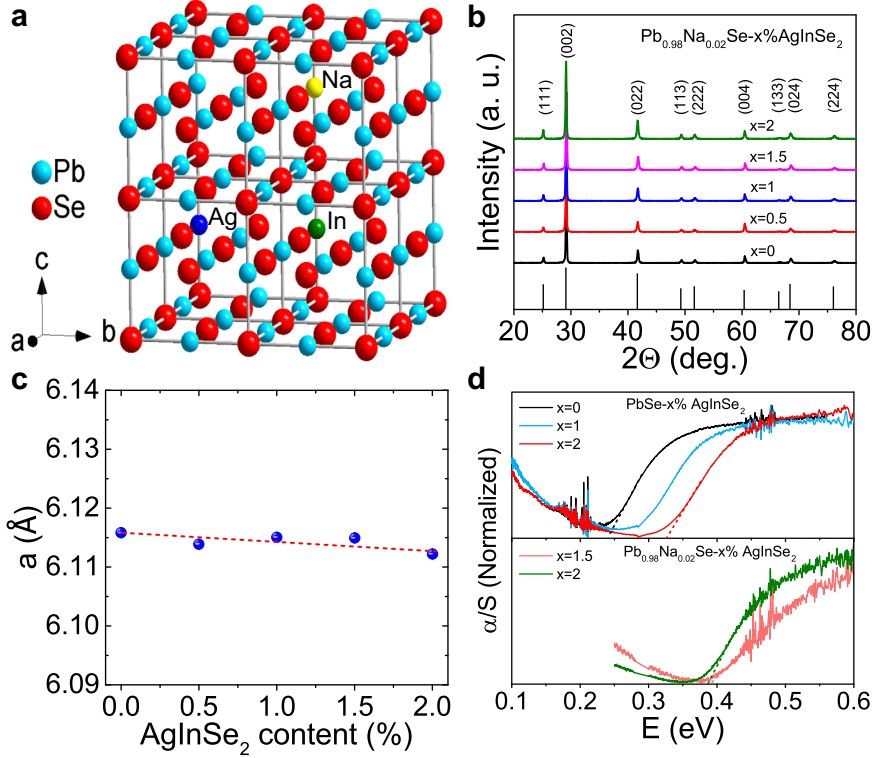

**Fig. 2 Crystal structure and bandgap. a** Schematic crystal structure of $Pb_{0.98}Na_{0.02}Se-x\%$ $AgInSe_2$ (LISS). **b** Powder XRD patterns of LISS. **c** Refined lattice constants of LISS. **d** Room temperature infrared spectra for $PbSe-x\%$ $AgInSe_2$ and $Pb_{0.98}Na_{0.02}Se-x\%$ $AgInSe_2$.

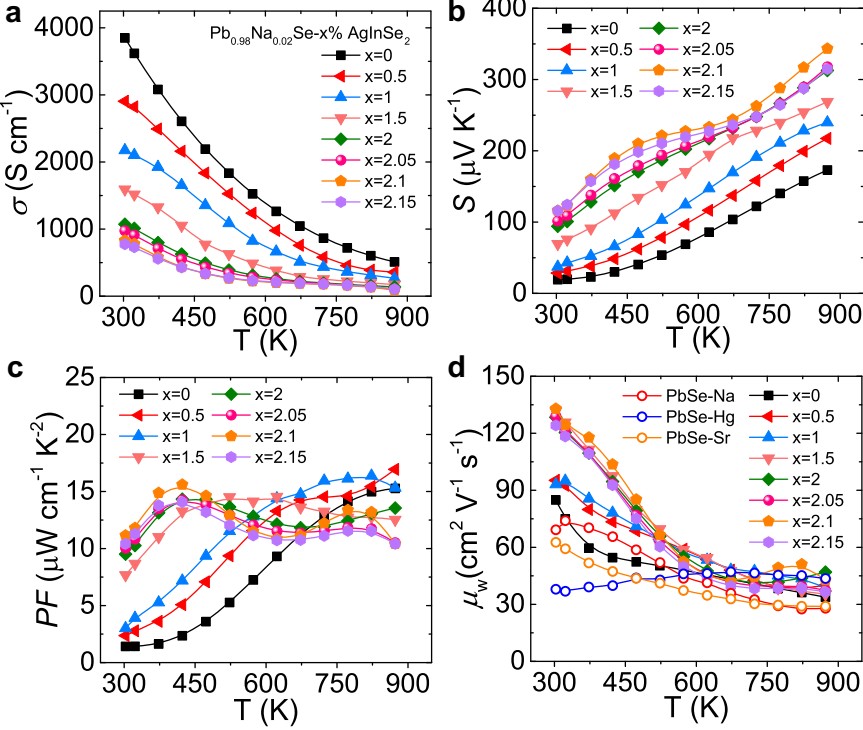

**Fig. 3 Electrical properties as a function of temperature for $Pb_{0.98}Na_{0.02}Se-x\%$ $AgInSe_2$ (LISS) compounds. a** Electrical conductivity. **b** Seebeck coefficient. **c** Power factor. **d** Weighted mobility. The hollow circles in **d** represent the weighted mobility of single-band and two-band PbSe-based materials.

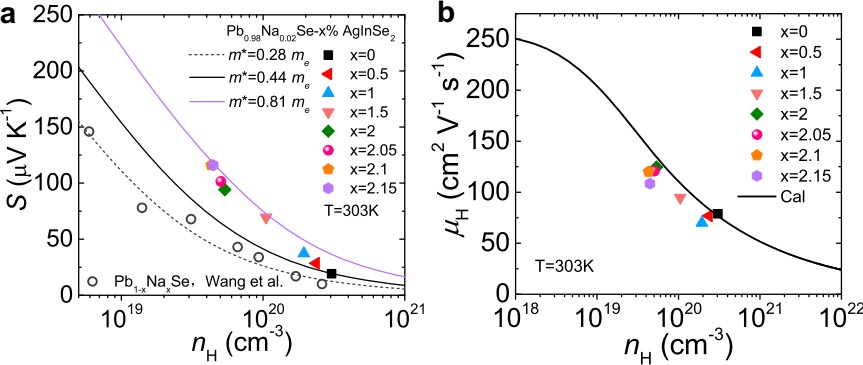

**Fig. 4 Pisarenko plot and Hall carrier mobility. a** Pisarenko plot of Seebeck coefficients as a function of Hall carrier concentration ($n_H$) for $Pb_{0.98}Na_{0.02}Se$ −$x$% AgInSe₂ (LISS). The solid black line is calculated assuming $m^* = 0.44m_e$ and the purple line represents the result assuming $m^* = 0.81m_e$ within the SPB model. The gray circles show the Pisarenko plot for Na-doped PbSe reported by Wang et al.[50]. **b** Hall carrier mobility ($\mu_H$) versus Hall carrier concentration ($n_H$) at 303 K.

whereas a much larger Seebeck coefficient value of 116 μV/K is achieved for $Pb_{0.98}Na_{0.02}Se−2.15\%$ AgInSe₂ sample. The dramatically promoted Seebeck coefficients will facilitate the enhancement of power factor (PF). Indeed, the PF have an apparent improvement especially at the 300–600 K temperature range for all doped samples (Fig. 3c). The room temperature PF value of $Pb_{0.98}Na_{0.02}Se$ is only ~1.4 μW cm⁻¹ K⁻². In sharp contrast, the room temperature PF increases to ~11.1 μW cm⁻¹ K⁻² when $x = 2.1$ and this value is continuously improved to ~15.6 μW cm⁻¹ K⁻² at 423 K (Fig. 3c).

To understand the nature of the improvement of Seebeck coefficient, the relationship of Seebeck coefficient as a function of carrier concentration (Pisarenko curve) was compared at room temperature (Fig. 4a). Generally, the Seebeck coefficient increases with decreasing carrier concentration. However, the Seebeck coefficient is largely departure from the theoretical values estimated by the single parabolic band (SPB) model, which indicates that a complex electronic band structure should be involved in the electrical transport properties. Accordingly, the effective mass ($m^*$) of LISS is largely increased from 0.44 $m_e$ to 0.81 $m_e$ with the introduction of AgInSe₂ (Fig. 4a and Supplementary Fig. 1b). In contrast, the effective mass of Na-doped PbSe is only ~0.28 $m_e$ (Fig. 4a). The Hall carrier mobility increases with doping and a maximum value of ~125 cm² V⁻¹ s⁻¹ is obtained when $x = 2$ (Fig. 4b), which is largely due to the depressed carrier concentration. Consequently, the weighted mobility ($\mu_W$) of LISS compounds is largely enhanced especially at the 300–600 K temperature range, which is higher than that of single-band and two-band PbSe-based materials (Fig. 3d).

DFT calculations were also conducted to understand the origin of the enhanced Seebeck coefficients. We observed significant change of the electronic band structure with the incorporation of AgInSe₂ in PbSe matrix (Fig. 5a). The bandgap is enlarged upon doping, which will depress the bipolar effect and facilitate the enhancement of Seebeck coefficient. These calculations are well in accordance with our experimental results. The experimental bandgap ($E_g$) is ~0.24 eV for the pristine PbSe, while the bandgap increases obviously with the incorporation of AgInSe₂ and a large bandgap ~0.33 eV is achieved for the PbSe−2% AgInSe₂ sample (Fig. 2d). The small bandgap of PbSe results from its unconventional chemical bonding mechanism (metavalent bonding). For a perfect half-filled p-band, the energy band structures resemble a metallic system. Yet, the bandgap opens due to a small Peierls distortion or charge transfer[41]. It is the charge transfer between Pb and Se that opens a small bandgap in PbSe given its perfect octahedral arrangements. DFT results show that the enlarged bandgap is mainly attributed to the incorporation of Ag. The eletronegativity difference between Ag and Se (~0.62) is larger than that between Pb

and Se (~0.22). Therefore, the substitution of Ag at Pb sites will strengthen the charge transfer between cation and anion, leading to an enlarged bandgap. Interestingly, the bandgap is further enlarged to ~0.38 eV with Na doping (Fig. 2d). In addition, the L band is flattened. The sharp peaks reflected in the density of states (DOS) for valence band also reveal the band flattening character (Fig. 5b). Simultaneously, the Σ band is elevated and hence the energy offset ($\Delta E_{1-2}$) between L and Σ band is shortened. Surprisingly, a third valence band at the Λ point is activated and it remains at the same energy level compared with the Σ band (Fig. 5a). These multiple valence bands enable large effective mass without significant affect the carrier mobility, which is the origin of enhanced Seebeck coefficient and the weighted mobility ($\mu_W$).

The electronic band structures of Ag and In doped PbSe were calculated (Supplementary Fig. 2a, b) to understand their roles in band manipulation. The Ag-doping and In-doping reflect p-type and n-type doping effect, respectively, which are consistent with previous experimental results[42,43]. Additionally, In-doping has a more important effect on decreasing energy offset ($\Delta E_{1-2}$) compared with the Ag-doping (Supplementary Fig. 2c), while Ag-doping plays a major role in enlarging the bandgap (Supplementary Fig. 2d). The orbital projected band structures reveal that the interaction between Pb-p and Se-p orbitals dominate the band structure (Supplementary Fig. 3a, b), which is consistent with previous study[44]. This is a typical feature of the metavalent bonding system[13,45,46]. The tight binding calculations reveal that the cation states have important effect on the shape of valence band although their orbital projections are not obvious[44]. Indeed, the Ag-d orbitals play an important role in modulating the third valence band Λ along Γ-L (Supplementary Fig. 3c). A similar phenomenon was also observed in Ag-Sr co-dope PbSe system[47]. In addition, the cation-site doping can also contribute to the conduction band (Supplementary Fig. 3d) depending on the nature of cation states. Our results indicate that Ag-In co-doping enable multiple valence band convergence, verifying that the cation-site doping is an effective way to modulate the valence band in PbSe. Similar effects can be expected in other materials, such as PbTe and GeTe, by employing the same chemical bonding mechanism as PbSe.

Using the lattice parameters extracted from the temperature-dependent synchrotron radiation x-ray diffraction (SR-XRD) patterns (Supplementary Fig. 4), we calculated the band structures as a function of temperature (Fig. 5c and Supplementary Fig. 5). Clearly, the bandgap increases with rising temperature, which is also verified experimentally (Fig. 5d). As revealed by Brod et al.[44], there is sufficient interaction between Pb-p and Te-p (Se-p in our case) to provide the molecular

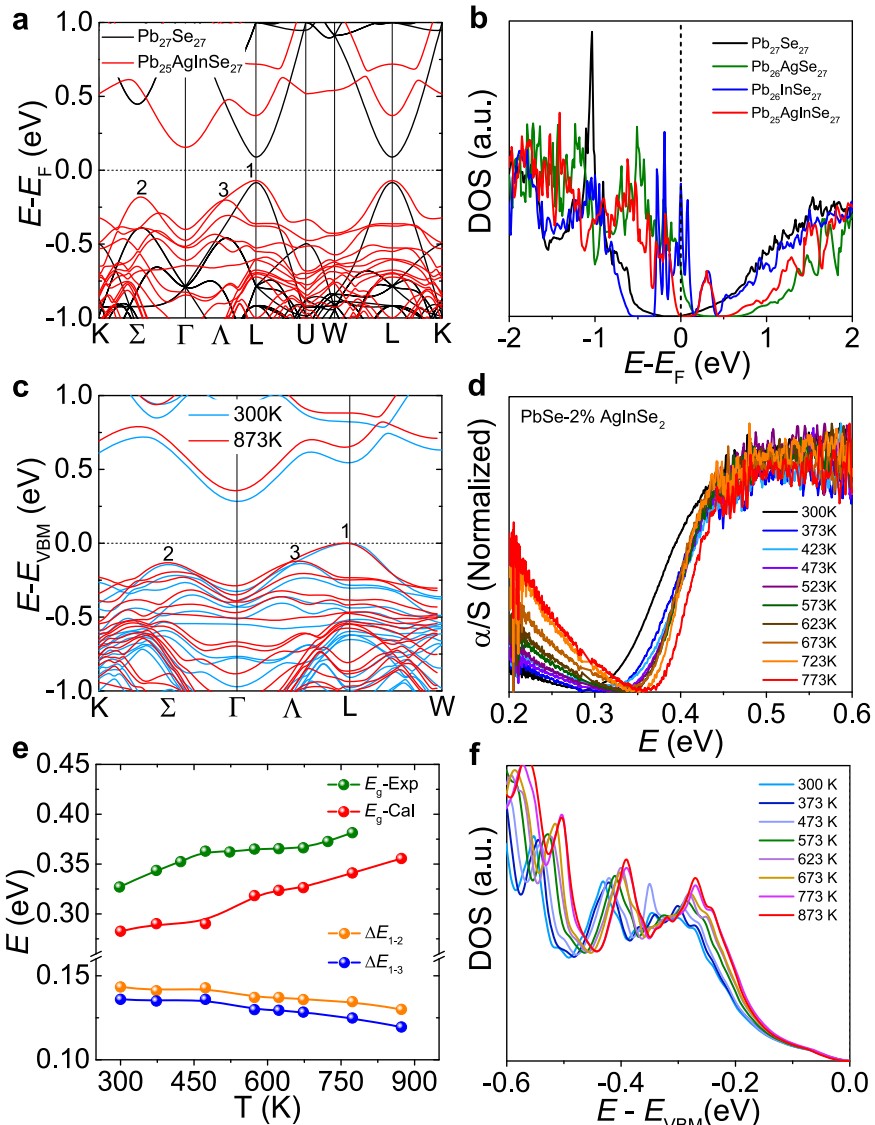

**Fig. 5 Electronic band structure. a** Electronic band structure of $Pb_{27}Se_{27}$ (black) and $Pb_{25}AgInSe_{27}$ (red). **b** Electronic density of states (DOS) near the Fermi level for $Pb_{27}Se_{27}$ (black), $Pb_{26}AgSe_{27}$ (green), $Pb_{26}InSe_{27}$ (blue) and $Pb_{25}AgInSe_{27}$ (red), respectively. **c** Electronic band structure of $Pb_{25}AgInSe_{27}$ at 300 K and 873 K, respectively. **d** Temperature-dependent infrared spectra of PbSe−2% $AgInSe_2$. **e** The experimental (green) and theoretical (red) bandgap ($E_g$) and the theoretical energy offset between VBM1 and VBM2 ($\Delta E_{1-2}$) and between VBM1 and VBM3 ($\Delta E_{1-3}$) as a function of temperature. **f** Temperature-dependent electronic DOS of $Pb_{25}AgInSe_{27}$ near the VBM.

orbitals with the proper s-type symmetry to place the VBM at L point. The weak s-p hybridization is a small addition to this effect. The thermal expansion will lead to a reduction of orbital overlap between p-orbitals[41] as well as a weakened s-p hybridization[48]. As a result, the energy of VBM (L point) in the electronic band structure decreases[49], resulting in an enlarged bandgap ($E_g$). The unobvious increase tendency of bandgap above ~500 K may be attributed to the band convergence, where the heavy valence bands dominate and the position of heavy valence bands are almost temperature independent[1]. The theoretical bandgap is smaller than the experimental result, which may be attribute to the neglect of the effect of thermal disorder on the bandgap in our calculations. Moreover, the energy offset ($\Delta E_{1-2}$) between L and $\Sigma$ bands decreases with increasing temperature (Fig. 5e). Interestingly, the energy offset ($\Delta E_{1-3}$) between L and $\Lambda$ also shows a decline tendency with rising temperature and its value is even smaller than $\Delta E_{1-2}$ in the whole temperature range (Fig. 5e). The convergence tendency and the involvement of the third valence

band is also reflected in the DOS corresponding to the valence band increases with increasing temperature (Fig. 5f). This convergence behavior is experimentally verified via Hall measurements, in which a maximum Hall coefficient ($R_H$) is observed (Supplementary Fig. 6a) and it is a sign of band convergence of the multi-valence bands[21,24]. Consequently, the effective mass ($m^*$) of $Pb_{0.98}Na_{0.02}Se$−2.05%$AgInSe_2$ increases from $0.73m_e$ to $2.16m_e$ with rising temperature, which is much higher than the $m^*$ of single Na-doped PbSe[50] (Supplementary Fig. 6b).

**Thermal transport properties and the dimensionless figure-of-merit ($ZT$).** Thermal conductivity is another important parameter for thermoelectric performance. The total thermal conductivity ($\kappa_{tot}$) decreases significantly with increasing $AgInSe_2$ (Fig. 6a). The $\kappa_{tot}$ is a sum of lattice thermal conductivity ($\kappa_L$) and electronic thermal conductivity ($\kappa_e$). The $\kappa_e$ was calculated by the Wiedemann-Franz relation, $\kappa_e = L\sigma T$, where $L$ (Supplementary Fig. 7a) is estimated by SPB model assuming acoustic phonon

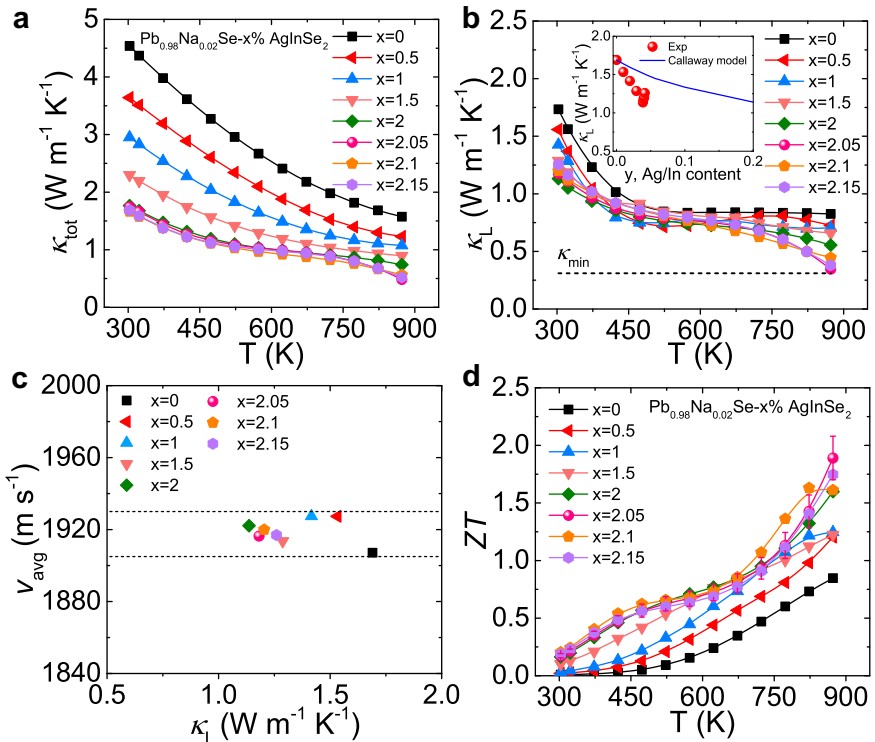

**Fig. 6 Thermal transport properties and dimensionless figure-of-merit *ZT* as a function of temperature for Pb$_{0.98}$Na$_{0.02}$Se−*x*% AgInSe$_2$ (LISS) compounds. a** Total thermal conductivity. **b** Lattice thermal conductivity. Inset shows the room-temperature lattice thermal conductivities departure from the theoretical line calculated by the Callaway model. **c** The average sound velocity ($v_{avg}$) versus lattice thermal conductivity ($\kappa_L$) for LISS compounds at room temperature. **d** Temperature-dependent *ZT* for LISS samples.

scattering dominates (Supplementary Fig. 6c). The $\kappa_e$ decreases remarkably with doping due to largely depressed electrical conductivity (Supplementary Fig. 7b). Furthermore, the $\kappa_L$ is obtained by subtracting the electronic contribution from the total thermal conductivity (Fig. 6b). Similarly, the $\kappa_L$ is largely suppressed with doping and the room temperature $\kappa_L$ values are much lower than the theoretical estimation by the Callaway model (Fig. 6b, inset). In addition, the $\kappa_L$ decreases with rising temperature and a clear departure from $T^{-1}$ relation is observed, demonstrating that the strong phonon scattering occurs. To further uncover the mechanism of the reduction of $\kappa_L$, sound velocities were measured for all the samples at room temperature (Supplementary Table 1). Interestingly, the average sound velocity ($v_{avg}$) slightly increases after doping (Fig. 6c). The deduced Grüneisen parameters ($\gamma$) and bulk modulus ($K$) of LISS have no obvious change (Supplementary Table 1). The lattice thermal conductivity can be expressed as $\kappa_L = \frac{1}{3}Cv_{avg}^2\tau$ based on the simple kinetic theory[51], where $C$ is the specific heat, $\tau$ is the phonon lifetime. Here, the $v_{avg}$ increases upon doping and thus the reduction of lattice thermal conductivity should be derived from the decrease of phonon lifetime. In another word, enhanced phonon scattering is the main origin of the largely suppressed lattice thermal conductivity.

Thanks to the complex band structure behavior and strong phonon scattering with the introduction of AgInSe$_2$, the *ZT* is largely enhanced in the whole temperature range and a maximum *ZT* value of ~1.9 is achieved at 873 K for Pb$_{0.98}$Na$_{0.02}$Se−2.05% AgInSe$_2$ sample (Fig. 6d). The high thermoelectric properties of LISS samples are reproducible (Supplementary Fig. 8).

**Microstructure and local structure analysis**. The TEM images of LISS sample display that abundant nanoscale precipitates are

embedded in PbSe matrix (Fig. 7a, b). In addition, strip-like dislocations are also observed (circled regions in Fig. 7a). Both nanoscale precipitates and multi-scale dislocations are effective phonon scattering centers[52]. The annular dark-field scanning transmission electron microscopy (STEM) image and the energy dispersive x-ray spectroscopy (EDS) elemental mappings exhibit obvious Ag-rich and In-rich patterns for the precipitates (Supplementary Fig. 9). Accordingly, the elemental distributions of Se and Na are relatively homogeneous in the entire area, whereas Pb-poor regions are observed within the precipitates (Supplementary Fig. 9). A clearer microstructural feature of the precipitate is revealed by performing HRTEM (Fig. 7c). The corresponding selected area electron diffraction pattern indicate a main cubic structure along [111], whereas the precipitates shows a different crystal structure from the PbSe matrix as another series of diffraction spots are exhibited which can be indexed to AgInSe$_2$ (Fig. 7d). The high angle annular dark field (HAADF) patterns show that the tetragonal AgInSe$_2$ is perfectly inserted in the cubic PbSe matrix as a nine-atom grid (Fig. 7e). We can also observe lattice dislocation in the HAADF (Fig. 7f). The lattice mismatch induced by precipitates and dislocations will introduce large strain fluctuations and thus enhance the phonon scattering[53]. Therefore, the lattice thermal conductivity of LISS was largely reduced to its amorphous limit of 0.31 Wm$^{-1}$ K$^{-1}$ at 873 K arising from the strong phonon scattering.

Understanding the atomic occupation of doping elements in the crystal lattice allow us elucidate their roles on manipulating the thermal or electrical transport properties. The XAFS spectroscopy is a powerful tool to investigate the local structure in materials[54–59]. Here, we performed XAFS measurements for PbSe, AgInSe$_2$ and LISS, respectively. The x-ray absorption near-edge structure (XANES) of Se *K*-edge and Pb *L$_3$*-edge do not show any change after the introduction of AgInSe$_2$ in PbSe

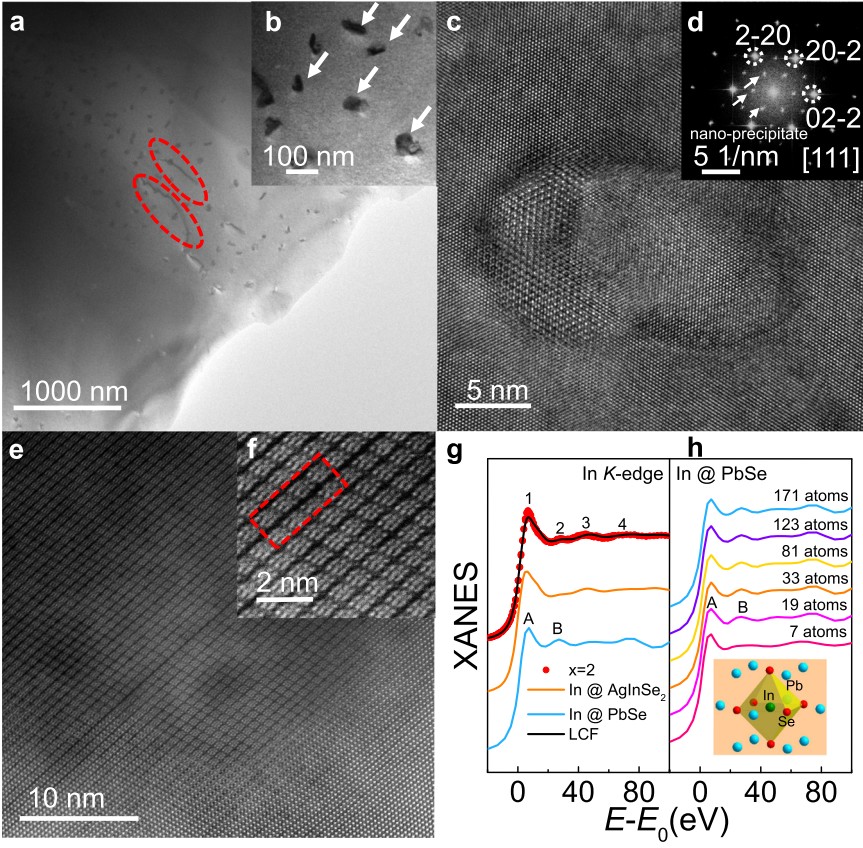

**Fig. 7 Microstructures and local structure analysis for high-performance LISS sample. a** Low magnification of bright-field TEM image for $Pb_{0.98}Na_{0.02}Se$ −2.05% $AgInSe_2$ sample. **b** The enlarged TEM pattern presents the nanoscale precipitates remarked by the arrows. **c** High resolution TEM (HRTEM) picture of a selected nanoprecipitate. **d** The corresponding selected area electron diffraction (SAED) pattern with cubic structure along [111]. **e, f** High angle annular dark field (HAADF) patterns for $Pb_{0.98}Na_{0.02}Se$−2.05% $AgInSe_2$. **g** Experimental XANES spectra of In $K$-edge for $Pb_{0.98}Na_{0.02}Se$−2% $AgInSe_2$ (red dots), and $AgInSe_2$ (orange line), respectively. The blue line shows the theoretical XANES spectrum of In $K$-edge for In-doped PbSe assuming that In occupies the Pb site. The black line represents a linear combination fitting (LCF) result of In $K$-edge of $Pb_{0.98}Na_{0.02}Se$−2% $AgInSe_2$ considering that the In $K$-edge of $AgInSe_2$ and In-doped PbSe serves as standards. **h** Multiple scattering calculations of In $K$-edge XANES for In-doped PbSe with different atomic clusters. The inset shows the nearest-two shell of In atom when it occupies the Pb site in PbSe matrix. The $E_0$ is the absorption edge energy of In $K$-edge of In foil.

(Supplementary Fig. 10), demonstrating a steady PbSe cubic matrix, which is well consistent with the XRD patterns. There are four main features in the XANES of In $K$-edge of LISS (Fig. 7g), in which 1, 3, 4 features can well reflected in the In $K$-edge of $AgInSe_2$, indicating that the most of In atoms may form $AgInSe_2$ in the system. Furthermore, we calculated the XANES spectra of In $K$-edge for the In-doped PbSe assuming that In occupies the Pb site, in which the peak B is well corresponding with the feature 2 of In $K$-edge of LISS. Moreover, after adding the second shell (12 Pb atoms) as shown by the 19-atoms cluster calculation (Fig. 7h), the feature B is well reflected and the calculation is almost convergent. Therefore, the origin of feature 2 is mainly arising from the multiple scattering of the photoelectrons by the second shell of Pb atoms in the PbSe matrix. The linear combination fitting (LCF) was used to evaluate the atomic occupied ratio of In atoms in each standards (Fig. 7g). Similar analysis was performed for the XANES of Ag $K$-edge (Supplementary Fig. 11). The LCF fitting results indicate that about 80% of Ag and In atoms form $AgInSe_2$ in the system (Supplementary Table 2), causing strong phonon scattering.

## Discussion

In summary, a combined effect of three activated valence bands and strong phonon scattering is realized via introducing $AgInSe_2$ in $Pb_{0.98}Na_{0.02}Se$ matrix. These multiple valence bands

convergence enable the enhancement of thermoelectric power factor at low temperature region and maintain at a high level at elevated temperature. Interestingly, local structure studies by XANES reveal that most of Ag or In atoms form $AgInSe_2$ secondary phase. The numerous nanoscale $AgInSe_2$ precipitates and multi-scale dislocations observed in the TEM will cause strong phonon scattering. Therefore, the lattice thermal conductivity is largely depressed. As a consequence, a distinguished dimensionless figure-of-merit $ZT$ of ~1.9 is achieved at 873 K, which is among the best bulk thermoelectric materials. This work proves that multiple valence bands could be activated in p-type PbSe and highlights the strong phonon scattering effect through the introduction of secondary phase with ultralow thermal conductivity, which guide a promising route to achieve excellent thermoelectric performance in bulk materials. The quantitative atomic occupation of doping elements provides a microscopic perspective to understand their role on manipulating transport properties. It is expected that more advanced thermoelectric materials can be achieved by employing the present strategy.

## Methods

**Synthesis**. High-purity starting materials, Pb (99.999%), Se (99.999%), Na (99.9%), Ag (99.99%), In (99.99 %) were weighted in stoichiometric ratio ($Pb_{0.98}Na_{0.02}Se$−$x$ % $AgInSe_2$) and loaded in carbon coating silica tubes under a $N_2$-filled glove box. The silica tubes were sealed under vacuum and then slowly heated to 1423 K in 24 h, soaked at this temperature for 10 h and followed by furnace cooling down to room temperature. The obtained ingots were grounded into powders and then

densified at 873 K for 6 min with a pressure of 50 MPa using spark plasma sintering (SPS-211Lx). Finally, highly dense bulk samples (>97% of theoretical density) were obtained.

**Thermoelectric property measurements**. The bulk samples were cut into rectangular solids ($3 \times 3 \times 10$ mm$^3$) and square pieces ($10 \times 10 \times 1$ mm$^3$) for electrical and thermal transport properties measurements, respectively. The Seebeck coefficients and electrical conductivities were measured using the Ulvac Riko ZEM-3 instrument. We calculated the thermal conductivity using the equation of $\kappa_{tot} = D \cdot C_p \cdot \rho$, where the thermal diffusivity ($D$) was determined using a laser flash method by the Netzsch LFA-457 facility, the heat capacity ($C_p$) was estimated using an empirical equation ($C_p/k_B$ atom$^{-1} = 3.07 + 4.7 \times 10^{-4}$ ($T$/K-300))[30], and the sample density is calculated by the dimensions and mass of the samples. The combined uncertainty of all measurements for determining the $ZT$ is less than 20%.

**Characterizations**. Room temperature powder XRD measurements were conducted using a D/MAX 2500 PC system with Cu K$_\alpha$ radiation. High-temperature SR-XRD patterns were performed for Pb$_{0.98}$Na$_{0.02}$Se−2% AgInSe$_2$ at the BL14B1 beamline of Shanghai synchrotron radiation facility. The wavelength of the x-ray is 0.6887 Å. The sample was heated from 300 K to 875 K at a rate of 5 K min$^{-1}$. The bandgap was measured using the Shimadzu Model UV-3600 Plus instrument and was estimated by the Kubelka-Munk equation. Pulse-echo method was used to measure the speed of sound and the waveforms were recorded using a Tektronix TBS 1102 oscilloscope. The Hall coefficient ($R_H$) was conducted by the Van der Pauw method using the Lake Shore 8400 Series. STEM and TEM were performed using a JEOL ARM200F equipped with cold field emission gun and ASCOR probe corrector. More details can be found in the Supporting Information.

**First-principles calculations**. Density functional theory calculations were performed using the projector-augmented wave method[60], as implemented in the Vienna Ab initio Simulation Package[61,62]. We utilized the revised Perdew-Burke-Ernzerhof[63] generalized gradient approximation to estimate the exchange-correlation interactions. A cutoff energy was set to 450 eV for the plane-wave expansion of the electron density and the Monkhorst-Pack $k$-point sampling 0.1 Å$^{-1}$ was used within all the calculations. The atomic positions were fully relaxed when the maximum residual ionic force and total energy difference are converged within 0.01 eV Å$^{-1}$ and $10^{-7}$ eV, respectively. Several $3 \times 3 \times 3$ supercells were constructed (Pb$_{27}$Se$_{27}$, Pb$_{26}$AgSe$_{27}$, Pb$_{26}$InSe$_{27}$, Pb$_{25}$AgInSe$_{27}$), avoiding the defect-defect interaction. The occupations of Ag or/and In atoms in the supercells were relaxed in our calculations. The temperature-dependent electronic band structures were performed using the experimental lattice parameters at elevated temperatures deriving from the SR-XRD data.

**X-ray absorption fine structure (XAFS) spectroscopy measurements**. The XAFS experiments were performed at BL01B1 beamline of Spring-8 in Japan. The electron energy of the storage ring is 8.0 GeV with a top-up filling of 99.5 mA accumulated current during the experiment. The Si (311) double-crystal monochromator was used for tuning the monochromatic beam. We measured the XAFS of Ag K-edge and In K-edge for AgInSe$_2$ in transmission mode. The XAFS measurements of Se K-edge, and Pb $L_3$-edge for PbSe and Pb$_{0.98}$Na$_{0.02}$Se−2% AgInSe$_2$ were conducted in transmission mode, while the measurements of Ag K-edge and In K-edge for Pb$_{0.98}$Na$_{0.02}$Se−2% AgInSe$_2$ were performed in fluorescence mode using 19-element Ge solid-state detector. All experimental XAFS spectra were preprocessed using the IFFEFIT package[64].

**XAFS calculation and analysis**. The x-ray absorption near-edge structure (XANES) calculations of Ag K-edge for Ag-doped PbSe and In K-edge for In-doped PbSe were performed based on the full multiple scattering (FMS) using FEFF9 program[65,66]. We use self-consistent field (SCF) method to estimate the atomic scattering potential. To investigate the doping site of Ag or In in PbSe, we simply replace the central Pb absorber with Ag or In atom while maintaining the coordinates. To achieve good convergence, the cluster radius for SCF and FMS was fixed as 8 and 10 Angstrom, respectively. LCF of In K-edge for Pb$_{0.98}$Na$_{0.02}$Se−2% AgInSe$_2$ was performed using the Athena software assuming that the XANES of In K-edge of AgInSe$_2$ and In-doped PbSe as the standards. A similar LCF analysis was applied for the XANES of Ag K-edge of Pb$_{0.98}$Na$_{0.02}$Se−2% AgInSe$_2$. Since we cannot ensure that the Ag or In atoms totally occupy the Pb site in PbSe matrix without formation of impurity phases, we thus used the calculated XANES as one of standards in the LCF analysis.

**Reporting summary**. Further information on research design is available in the Nature Research Reporting Summary linked to this article.

## Data availability

The authors declare that the data supporting the findings of this study are available within the paper and its Supplementary Information files. All of the other data are available from the authors upon reasonable request.

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

## Acknowledgements

The authors thank BL14B1 at Shanghai synchrotron radiation facility (SSRF) for the SR-XRD measurements. We thank BL01B1 at Spring-8 for the XAFS experiments (Proposal Number: 2021B1109). This work was supported by National Key Research and Development Program of China (2018YFA0702100 and 2018YFB0703600), National Natural Science Foundation of China (51772012, 52002042 and 52002011), National Postdoctoral Program for Innovative Talents (BX20200028), the Beijing Natural Science Foundation (JQ18004), and 111 Project (B17002). L.-D.Z. appreciates the support of the high-performance computing (HPC) resources at Beihang University, the National Science Fund for Distinguished Young Scholars (51925101), and center for High Pressure Science and Technology Advanced Research (HPSTAR) for TEM measurements.

## Author contributions

Y.Z. and L.-D.Z. prepared the samples, carried out the experiments, analyzed the results and wrote the paper. T.H. and X.G. performed the TEM experiments. D.W. carried out the DFT calculations. Y.Z., L.H. and T.I. conducted the XAFS measurements and analyzed the data. S.Z., B.Q., H.S. and L.S. performed the SR-XRD experiments. All authors coedited the manuscript.

## Competing interests

The authors declare no competing interests.
