## [Peer Review File · Nature Communications]

Multiple valence bands convergence and strong phonon scattering lead to high thermoelectric performance in p-type PbSeREVIEWER COMMENTS

Reviewer #1 (Remarks to the Author):

Band convergence is an effective way to improve the electrical properties of thermoelectric materials. Here, the authors reported that multiple valence bands could be activated in p-type PbSe-AgInSe₂ system, which is a novel point compared with the traditional two-band convergence. These results will motivate researchers to find more strategies on band structure engineering. AgInSe₂ has an intrinsic low lattice thermal conductivity. The introduction of AgInSe₂ leads to strong phonon scattering verified by the nano-scale precipitates and dislocations. As a result, a large ZT (~2.1) was achieved in this work and it shows a good reproducible thermoelectric performance. The paper reported an extensive study on p-type PbSe-AgInSe₂ and it is well written. Therefore, I recommend the paper to be published in Nature communications. Suggestions or comments are given below.

1. The microstructure study didn't give a quantitative results on the occupation of Ag and In atoms in this system. So, the sentence -"Local structure and microstructure analysis reveal that about 80 percent of Ag and In atoms form AgInSe₂ as nano-scale precipitates" -in the Abstract may make misunderstanding, which should be modified.
2. The temperature-dependent of band-gap may be another evidence for the band convergence. The increase tendency of band gap become unobvious above ~ 500K, which indicate that the heavy valence bands may dominates above this temperature. This phenomenon is also consistent with the temperature-dependent Hall measurement that RH peak appears around 500K (Figure S5).
3. Mechanical property is important for the transport performance of materials. What is the effect on the mechanical property of PbSe with the introduction of AgInSe₂?
4. The Lorenz number was calculated assuming acoustic-phonon scattering dominates. How to prove this?
5. What is the uncertainty for the thermoelectric properties measurements?

Reviewer #2 (Remarks to the Author):

In this manuscript, the authors investigated the thermoelectric properties of Pb_{0.98}Na_{0.02}Se_{1-x}AgInSe₂ (LISS). An exceptional figure-of-merit ZT of ~2.1 at 873K was achieved. They performed a systematic study on the electrical/thermal transport properties and microstructures for this system. The introduction of AgInSe₂ enlarges the band gap, suppressing the bipolar effect. It is very important to achieve high thermoelectric performance at high temperature regime, but has been challenging to obtain. Interestingly, the incorporation of AgInSe₂ facilitates the convergence of multiple valence bands, resulting in high weighted mobility and large power factor. The presence of nano-scale AgInSe₂ precipitates and dislocations results in strong phonon scattering. As a result, a combination of band convergence and strong phonon scattering gives record-high thermoelectric performance in PbSe. The local structure analysis by XAFS is very interesting, providing a microscopic perspective to understand the role of doped elements. This is a very solid work and it is suitable for the scope of Nature Communications. This work shows that PbSe thermoelectrics can compete with the much expensive PbTe. It is highly recommended to be published in Nature Communications after addressing minor points as given below:

1. In the "Introduction" section, please cite related works when you mentioned the quality factor B (The lattice thermal conductivity is another important parameter for the thermoelectric performance indicated by the quality factor B).
2. It is mentioned that "the tetragonal AgInSe₂ is perfectly inserted to the PbSe matrix as nano-scale

precipitates revealed by the transmission electron microscopy". This sentence can make readers confused because a part of Ag and In atoms occupies Pb sites.

3. The effective masses increase with the introduction of AgInSe₂ as indicated by the Pisarenko plot. Please show the effective masses for each sample, which is more straightforward for the readers.

4. The measurement of sound velocities is not mentioned in the experimental section.

5. The heat capacity (C_p) calculated by the Dulong-Petit law will underestimate the thermal conductivity at high temperature. Estimating the C_p by the empirical equation is more accurate. It is not necessary to use the C_p value estimated by the Dulong-Petit law.

Reviewer #3 (Remarks to the Author):

see separate file

The present manuscript provides interesting data describing the thermoelectric properties of PbSe doped with AgInSe₂. This doping / alloying leads to a reduction in the thermal conductivity and an improvement of band convergence enabling a zT value slightly above 2. Thermoelectric materials are considered a viable option to improve energy conversion since they convert waste heat into electrical energy or enable efficient cooling. Hence identifying promising thermoelectric materials is a timely topic, which could be suitable for Nature Communications. Nevertheless, there are several reasons why the present manuscript does not meet my expectations for a manuscript to be published in Nature Communications.

One of the main claims is that there is multi-band convergence. Yet, the nature of these bands and how they could possibly be described and explained is missing. One of the main claims of the present manuscript is the idea that more than 3 bands can contribute, yet the proof for this claim and the explanation of the nature of these bands appears rather incomplete. I would like to see very strong and convincing evidence that indeed more than two bands contribute and what their nature is. To mention one option to provide such evidence: several groups have recently employed tight-binding methods to explain the band structure of related chalcogenides [1,2]. Such calculations could be performed to explore the potential nature and more importantly origin of the bands involved. I am not aware of any study that has claimed and proven so far that three bands can provide a contribution to the thermoelectric performance of PbSe.

Then, I am also concerned about the apparent disagreement between theory and experiment. The authors claim that their experiment shows an increase of band gap upon alloying with AgInSe₂. Yet, the DFT calculations presented in fig. 5b do not seem to support this conclusion.

Finally, in recent years the thermoelectric properties in lead chalcogenides have been discussed in terms of the underlying bonding mechanism, which must be related to the corresponding band structure [1,3]. A discussion of the fundamental bonding mechanism relevant here is missing. Such a discussion is important since it can help to predict and explain which materials and changes of bonding can improve the performance of a given thermoelectric material.

[1] Chemistry of Materials 32 (22), 9771-9779 (2020)

[2] Advanced Materials 30, 1801787 (2018)

[3] Advanced Materials 32, 202005533 (2020)

Reviewer #1 (Remarks to the Author):

General comment:

Band convergence is an effective way to improve the electrical properties of thermoelectric materials. Here, the authors reported that multiple valence bands could be activated in p-type PbSe-AgInSe₂ system, which is a novel point compared with the traditional two-band convergence. These results will motivate researchers to find more strategies on band structure engineering. AgInSe₂ has an intrinsic low lattice thermal conductivity. The introduction of AgInSe₂ leads to strong phonon scattering verified by the nano-scale precipitates and dislocations. As a result, a large ZT (~2.1) was achieved in this work and it shows a good reproducible thermoelectric performance. The paper reported an extensive study on p-type PbSe-AgInSe₂ and it is well written. Therefore, I recommend the paper to be published in Nature Communications. Suggestions or comments are given below.

Response: We appreciate the reviewer 1 for his/her solid summary and affirmation for our work. Your insightful comments will strengthen our work.

Comment 1: The microstructure study didn't give a quantitative results on the occupation of Ag and In atoms in this system. So, the sentence -“Local structure and microstructure analysis reveal that about 80 percent of Ag and In atoms form AgInSe₂ as nano-scale precipitates” -in the Abstract may make misunderstanding, which should be modified.

Response: Thanks for your good suggestions. We have made a revision for this sentence. In addition, we made a change to the “abstract” to meet the requirement of Nature Communications (150 words or fewer).

Revision: Abundant nano-scale precipitates and dislocations result in strong phonon scattering and thus ultralow lattice thermal conductivity. Consequently, we achieve an exceptional ZT of ~ 1.9 at 873 K in p-type PbSe. This work demonstrates that a combination of band manipulation and microstructure engineering can be realized by tuning the composition, which is expected to be a general strategy for improving the thermoelectric performance in bulk materials.

Comment 2: The temperature-dependent of band-gap may be another evidence for the band convergence. The increase tendency of band gap become unobvious above ~ 500K, which indicate that the heavy valence bands may dominates above this temperature. This phenomenon is also consistent with the temperature-dependent Hall measurement that R_H peak appears around 500K (Figure S5).

Response: It is a good point. A related discussion has been added in our manuscript.

Revision: The unobvious increase tendency of bandgap above ~ 500K may be attributed to the band convergence, where the heavy valence bands dominate and the

position of heavy valence bands are almost temperature independent¹.

1. Pei, Y. *et al.* Convergence of electronic bands for high performance bulk thermoelectrics. *Nature* **473**, 66-69 (2011).

Comment 3: Mechanical property is important for the transport performance of materials. What is the effect on the mechanical property of PbSe with the introduction of AgInSe₂?

Response: Thanks for your comments. We calculated the bulk modulus (*K*) as shown in Table S1. There is no significant effect on the mechanical property of PbSe with introducing AgInSe₂ because the bulk modulus has no obvious change.

Revision: The deduced Grüneisen parameters (γ) and bulk modulus (*K*) of LISS have no obvious change (Table S1).

Supplementary Table 1. Various parameters (longitudinal sound velocity (v_l), transverse sound velocity (v_t), average sound velocity (v_{avg}), Poisson ratio (ν_p), Grüneisen parameter (γ), and bulk modulus (*K*)) of Pb_{0.98}Na_{0.02}Se - *x*% AgInSe₂. The Poisson ratio (ν_p) is calculated by $\nu_p = \frac{1-2(v_t/v_l)^2}{2-2(v_t/v_l)^2}$, the Grüneisen parameter (γ) is obtained using $\gamma = \frac{3}{2} \left(\frac{1+\nu_p}{2-3\nu_p} \right)$ and the bulk modulus (*K*) is given by $K = \rho \left(v_l^2 - \frac{4}{3} v_t^2 \right)$ (ρ is the density of sample).

Sample	v_l (m/s)	v_t (m/s)	v_{avg} (m/s)	ν_p	γ	K (GPa)
x =0	3165.6	1708.6	1907.1	0.294	1.74	48.9
x =0.5	3192.9	1726.9	1927.3	0.293	1.73	50.2
x =1	3214.7	1726.1	1927.4	0.297	1.75	51.2
x =1.5	3151.7	1715.4	1913.5	0.289	1.71	48.0
x =2	3217.4	1720.9	1922.1	0.299	1.77	50.9
x =2.05	3148.7	1718.3	1916.4	0.288	1.70	47.6
x =2.1	3179.8	1720.3	1919.9	0.293	1.73	48.3
x =2.15	3149.7	1718.8	1916.9	0.288	1.70	47.4

Comment 4: The Lorenz number was calculated assuming acoustic-phonon scattering dominates. How to prove this?

Response: Thanks for your comments. The carrier scattering mechanism can be revealed in the log (μ_H)-log (T) relation. A log (μ_H)-log (T) plot displays T^{-3/2} behavior for *x* = 2.05 sample when T < 600 K (see below Figure), demonstrating that acoustic-phonon scattering dominates. The deviation from T^{-3/2} relation at high temperature is due to the increase of effective mass.

Revision: The κ_e was calculated by the Wiedemann-Franz relation, $\kappa_e = L\sigma T$, where L (Figure S7a) is estimated by SPB model assuming acoustic phonon scattering dominates (Figure S6c).

Supplementary Figure 6. (a) Temperature dependence of Hall coefficient (R_H) of $\text{Pb}_{0.98}\text{Na}_{0.02}\text{Se}-2.05\% \text{AgInSe}_2$. (b) Effective mass as a function of temperature for $\text{Pb}_{0.98}\text{Na}_{0.02}\text{Se}-2.05\% \text{AgInSe}_2$ and $\text{Pb}_{1-x}\text{Na}_x\text{Se}$. (c) Temperature-dependent Hall mobility of $\text{Pb}_{0.98}\text{Na}_{0.02}\text{Se} - 2.05\% \text{AgInSe}_2$, which displays a $T^{-3/2}$ behavior when $T < 600$ K, demonstrating that acoustic-phonon scattering dominates. The deviation from $T^{-3/2}$ relation at high temperature is due to the increase of effective mass.

Comment 5: What is the uncertainty for the thermoelectric properties measurements?

Response: Thanks for your comments. The standard samples were measured and these measurement data are compared with the reference value. The uncertainties of thermoelectric parameters are given below. In Figure 1 and 2, the measurements of thermal diffusivity (D), electrical resistivity (ρ) and Seebeck coefficient (S) are well consistent with their reference values. The error of thermal diffusivity is less than 2% (Figure 1). The uncertainties of electrical resistivity (ρ) and Seebeck coefficient (S) are less than 2% and 5%, respectively (Figure 2). The combined uncertainty of all measurements for determining the ZT is less than 20%.

Figure 1. (a) Comparison of thermal diffusivity (D) between measurement and reference for standard sample (Inconel). (b) The uncertainty of D .

Figure 2. Comparison of (a) electrical resistivity (ρ) and (c) Seebeck coefficient (S) between measurement and standard sample (Constantan), respectively. The corresponding uncertainties are shown in (b) and (d).

Reviewer #2 (Remarks to the Author):

General comment:

In this manuscript, the authors investigated the thermoelectric properties of $\text{Pb}_{0.98}\text{Na}_{0.02}\text{Se-x}\% \text{AgInSe}_2$ (LISS). An exceptional figure-of-merit ZT of ~ 2.1 at 873K was achieved. They performed a systematic study on the electrical/thermal transport properties and microstructures for this system. The introduction of AgInSe_2 enlarges the band gap, suppressing the bipolar effect. It is very important to achieve high thermoelectric performance at high temperature regime, but has been challenging to obtain. Interestingly, the incorporation of AgInSe_2 facilitates the convergence of multiple valence bands, resulting in high weighted mobility and large power factor. The presence of nano-scale AgInSe_2 precipitates and dislocations results in strong phonon scattering. As a result, a combination of band convergence and strong phonon scattering gives record-high thermoelectric performance in PbSe. The local structure analysis by XAFS is very interesting, providing a microscopic perspective to understand the role of doped elements. This is a very solid work and it is suitable for the scope of Nature Communications. This work shows that PbSe thermoelectrics can compete with the much expensive PbTe. It is highly recommended to be published in Nature Communications after addressing minor points as given below:

Response: We thank the reviewer 2 for his/her positive comments and valuable suggestions, which is a publishable justification for our submission.

Comment 1: In the “Introduction” section, please cite related works when you mentioned the quality factor B (The lattice thermal conductivity is another important parameter for the thermoelectric performance indicated by the quality factor B).

Response: Thanks for your comments. References about quality factor B are cited in the sentence.

Revision: The lattice thermal conductivity is another important parameter for the thermoelectric performance indicated by the quality factor B ($B \propto \mu\omega/\kappa_L$)^{30, 31}.

30. Kang, S. D., Snyder G. J. Transport property analysis method for thermoelectric materials material: quality factor and the effective mass model. arXiv:1710.06896 [cond-mat.mtrl-sci] (2017).

31. Tan, G., Zhao, L.-D. & Kanatzidis, M. G. Rationally Designing High-Performance Bulk Thermoelectric Materials. *Chem. Rev.* **116**, 12123-12149 (2016).

Comment 2: It is mentioned that “the tetragonal AgInSe_2 is perfectly inserted to the PbSe matrix as nano-scale precipitates revealed by the transmission electron microscopy”. This sentence can make readers confused because a part of Ag and In atoms occupies Pb sites.

Response: We appreciate your comments. This sentence has been modified.

Revision: The sentence mentioned above is changed to “Nano-scale AgInSe₂ precipitates are revealed by the transmission electron microscopy (TEM).”

Comment 3: The effective masses increase with the introduction of AgInSe₂ as indicated by the Pisarenko plot. Please show the effective masses for each sample, which is more straightforward for the readers.

Response: Thanks for your good suggestions. The effective masses as a function of AgInSe₂ content are given in our manuscript.

Revision: the effective mass (m^*) of LISS is largely increased from $0.44 m_e$ to $0.81 m_e$ with the introduction of AgInSe₂ (Figure 4a, Figure S1b).

Supplementary Figure 1. (a) Hall carrier concentrations and (b) density-of-states effective mass of Pb_{0.98}Na_{0.02}Se - x%AgInSe₂ (LISS) with increasing AgInSe₂ content at 303K.

Comment 4: The measurement of sound velocities is not mentioned in the experimental section.

Response: Thanks for your comments. The measurements of sound velocity have added in the experimental section.

Revision: Pulse-echo method was used to measure the speed of sound and the waveforms were recorded using a Tektronix TBS 1102 oscilloscope.

Comment 5: The heat capacity (C_p) calculated by the Dulong-Petit law will underestimate the thermal conductivity at high temperature. Estimating the C_p by the empirical equation is more accurate. It is not necessary to use the C_p value estimated by the Dulong-Petit law.

Response: We appreciate your good suggestions. We estimated the heat capacity using the empirical equation for all samples. The total thermal conductivity (κ_{tot}), lattice thermal conductivity (κ_L) and the figure-of-merit ZT have been recalculated.

Revision: we recalculated the total thermal conductivity (κ_{tot}), lattice thermal conductivity (κ_L) and the dimensionless figure-of-merit ZT . Accordingly, Figure 1, Figure 6 and Figure S8 have been revised.

Fig. 1 Multiple valence bands enable high ZT values in p-type PbSe. **a** Schematic diagram of multi-bands (L, Σ , Λ) involvement in transport. The Brillouin zone shows that the degeneracies at the L, Σ , and Λ points are 4, 12, and 8, respectively. **b** The activated third band Λ enables higher ZT values compared with the single-band and two-band PbSe-based materials.

Fig. 6 Thermal transport properties and dimensionless figure-of-merit ZT as a function of temperature for $\text{Pb}_{0.98}\text{Na}_{0.02}\text{Se}-x\% \text{AgInSe}_2$ (LISS) compounds. **a Total thermal conductivity. **b** Lattice thermal conductivity. Inset shows the room-temperature lattice thermal conductivities departure from the theoretical line calculated by the Callaway model. **c** The average sound velocity (v_{avg}) versus lattice thermal conductivity (κ_L) for LISS compounds at room temperature. **d** Temperature-dependent ZT for LISS samples.**

Supplementary Figure 8. Temperature-dependent (a) electrical conductivity, (b) Seebeck coefficient, (c) total thermal conductivity, and (d) dimensionless figure-of-merit ZT for several $x=2.05$ and $x=2.1$ samples, respectively.

Reviewer #3 (Remarks to the Author):

General comment:

The present manuscript provides interesting data describing the thermoelectric properties of PbSe doped with AgInSe₂. This doping / alloying leads to a reduction in the thermal conductivity and an improvement of band convergence enabling a zT value slightly above 2. Thermoelectric materials are considered a viable option to improve energy conversion since they convert waste heat into electrical energy or enable efficient cooling. Hence identifying promising thermoelectric materials is a timely topic, which could be suitable for Nature Communications. Nevertheless, there are several reasons why the present manuscript does not meet my expectations for a manuscript to be published in Nature Communications.

Response: We appreciate your valuable comments and suggestions, which will strengthen our work. Hopefully, our revised manuscript could meet your expectations.

Comment 1: One of the main claims is that there is multi-band convergence. Yet, the nature of these band and how they could possibly be described and explained is missing. One of the main claims of the present manuscript is the idea that more than 3 bands can contribute, yet the proof for this claim and the explanation of the nature of these bands appears rather incomplete. I would like to see very strong and convincing evidence that indeed more than two bands contribute and what their nature is. To mention one option to provide such evidence: several groups have recently employed tight-binding methods to explain the band structure of related chalcogenides [1,2]. Such calculations could be performed to explore the potential nature and more importantly origin of the bands involved. I am not aware of any study that has claimed and proven so far that three bands can provide a contribution to the thermoelectric performance of PbSe.

[1] Chemistry of Materials 32 (22), 9771-9779 (2020)

[2] Advanced Materials 30, 1801787 (2018)

Response: Thanks for your insightful comments. Our DFT calculations reveal that a third valence band Λ along Γ -L is activated. The large weighted mobility and effective mass also reflect the multi-band convergence indirectly. A comparison of the density-of-states effective mass for various p-type PbSe-based materials are shown in the Table 1 below.

Indeed, the tight-binding methods are powerful tool to understand the nature of electronic band structure. However, it is hard to employ tight-binding calculations for Ag-In co-doped PbSe since the supercells contain too many atoms. Instead, we calculated the atomic orbital projected band structure by DFT to understand their nature. The conduction and valence bands of PbSe system are dominated by the Pb-p and Se-p states, respectively, which is in line with the tight-binding calculations for PbTe [Brod, M. K., et al. *Chem. Mater.* **32**, 9771-9779 (2020)]. Owing to the rock-salt structure of PbSe, these p-bands are half-filled forming a σ -bond, which is

characteristic of metavalent bonding [Wuttig, M., et al. *Adv. Mater.* **30**, e1803777 (2018)]. Similar to PbTe, the valence band maximum (L band) of PbSe is contributed by the p-states. The projected electronic band structure also implies that the third valence band Λ show a large contribution by the Ag 4d state and Se 4p state. This may explain the promoted band convergence in PbSe by alloying with AgInSe₂.

A similar electronic band structure can also be found in Ag-Sr co-doped PbSe system [Luo, Z. Z. *et al.* *Angew. Chem.* 2021, 133, 272 – 277]. As shown in Figure 2 below, the valence band 2 (Σ) and valence band 3 (Λ) are almost at the same energy level. The energy offset between valence band 1 (L) and the other two valence bands (Σ and Λ) is ~ 0.17 eV. They found a strong band convergence behavior in this system. However, they didn't mention the underlying multi-band convergence behavior.

Table 1. Density-of-states effective masses (m^*) for various p-type PbSe-based materials.

sample	m^* (m_e) at 300K	m^* (m_e) at 773K	reference
PbSe-Na-Ag-In	0.81	2.16	This work
PbSe-Na	0.28	0.7	1
PbSe-Ag-Sr	0.4	1.1	2
PbSe-Ag-Ba	0.4	1.0	2
PbSe-Na-Hg	0.45	1.3	3
PbSe-Cd-Na-Te	0.57		4
PbSe-Ag	0.35		5
PbSe-Na-Ca	0.56		6
PbSe-Na-Ba	0.56		6
PbSe-Na-Sr	0.48		6

1 Wang, H., Pei, Y., LaLonde, A. D. & Snyder, G. J. Heavily doped p-type PbSe with high thermoelectric performance: an alternative for PbTe. *Adv. Mater.* **23**, 1366-1370 (2011).

2 Luo, Z. Z. *et al.* Strong Valence Band Convergence to Enhance Thermoelectric Performance in PbSe with Two Chemically Independent Controls. *Angew. Chem. Int. Ed.* **60**, 268-273 (2021).

3 Hodges, J. M. *et al.* Chemical Insights into PbSe- x%HgSe: High Power Factor and Improved Thermoelectric Performance by Alloying with Discordant Atoms. *J. Am. Chem. Soc.* **140**, 18115-18123 (2018).

4 Tan, G., Zhao, L.-D. & Kanatzidis, M. G. Rationally Designing High-Performance Bulk Thermoelectric Materials. *Chem. Rev.* **116**, 12123-12149 (2016).

5 Wang, S. *et al.* Exploring the doping effects of Ag in p-type PbSe compounds with enhanced thermoelectric performance. *J. Phys. D: Appl. Phys.* **44**, 475304 (2011).

6 Lee, Y. *et al.* High-performance tellurium-free thermoelectrics: all-scale

hierarchical structuring of p-type PbSe-MSe systems (M = Ca, Sr, Ba). *J. Am. Chem. Soc.* **135**, 5152-5160 (2013).

Figure 2 in reference [Luo, Z. Z. *et al.* *Angew. Chem.* 2021, 133, 272 – 277]. Electronic band structures and density-of-states (DOS) for Ag-doped PbSe (a, b), Ag-doped and SrSe-alloyed PbSe (c, d), and Ag-doped and BaSe-alloyed PbSe (e, f).

Revision: The electronic band structures of Ag and In doped PbSe were calculated (Figure S2a, S2b) to understand their role in band manipulation. The Ag-doping and In-doping reflect p-type and n-type doping effect, respectively, which are consistent with previous experimental results^{42,43}. Additionally, In-doping has a more important effect on decreasing energy offset (ΔE_{1-2}) compared with the Ag-doping (Figure S2c), while Ag-doping plays a major role in enlarging the bandgap (Figure S2d). The orbital projected band structures reveal that the interaction between Pb-p and Se-p orbitals dominate the band structure (Figure S3a, S3b), which is consistent with previous study⁴⁴. This is a typical feature of the metavalent bonding system^{13,45,46}. The tight binding calculations reveal that the cation states have important effect on the shape of valence band although their orbital projections are not obvious⁴⁴. Indeed, the Ag-d orbitals play an important role in modulating the third valence band Λ along Γ -L (Figure S3c). A similar phenomenon was also observed in Ag-Sr co-dope PbSe system⁴⁷. In addition, the cation-site doping can also contribute to the conduction band (Figure S3d) depending on the nature of cation states. Our results indicate that Ag-In co-doping enable multiple valence band convergence, verifying that the cation-site doping is an effective way to modulate the valence band in PbSe. Similar effects can be expected in other materials, such as PbTe and GeTe, by employing the same chemical bonding mechanism as PbSe.

13. Wuttig, M., Deringer, V. L., Gonze, X., Bichara, C. & Raty, J. Y. Incipient metals: functional materials with a unique bonding mechanism. *Adv. Mater.* **30**, e1803777 (2018).
42. Wang, S. *et al.* Exploring the doping effects of Ag in p-type PbSe compounds with enhanced thermoelectric performance. *J. Phys. D: Appl. Phys.* **44**, 475304 (2011).
43. Androulakis, J., Lee, Y., Todorov, I., Chung, D.-Y. & Kanatzidis, M. High-temperature thermoelectric properties of n-type PbSe doped with Ga, In, and Pb. *Phys. Rev. B* **83** (2011).
44. Brod, M. K., Toriyama, M. Y. & Snyder, G. J. Orbital chemistry that leads to high valley degeneracy in PbTe. *Chem. Mater.* **32**, 9771-9779 (2020).
45. Maier, S. *et al.* Discovering electron-transfer-driven changes in chemical bonding in lead chalcogenides (PbX, where X = Te, Se, S, O). *Adv. Mater.* **32**, e2005533 (2020).
46. Raty, J. Y. *et al.* A quantum-mechanical map for bonding and properties in solids. *Adv. Mater.* **31**, e1806280 (2019).
47. Luo, Z. Z. *et al.* Strong valence band convergence to enhance thermoelectric performance in PbSe with two chemically independent controls. *Angew. Chem. Int. Ed.* **60**, 268-273 (2021).

Supplementary Figure 2. Electronic band structures of (a) $\text{Pb}_{26}\text{AgSe}_{27}$ and (b) $\text{Pb}_{26}\text{InSe}_{27}$. (c) The energy offset (ΔE_{1-2}) between L and Σ valence band. (d) Theoretical bandgaps (E_g) for pristine, Ag-doped, In-doped and Ag-In co-doped PbSe.

Supplementary Figure 3. The atomic orbital projected band structure of $\text{Pb}_{25}\text{AgInSe}_{27}$. (a) The conduction band is dominated by Pb-p orbitals, while the valence band contain considerable Pb-s character. (b) The Se-p orbital primarily contributes to the valence band. (c) The Ag-d orbitals have a considerable contribution to the valence band. (d) There is distinct In-s character at the conduction band.

Comment 2: Then, I am also concerned about the apparent disagreement between theory and experiment. The authors claim that their experiment shows an increase of band gap upon alloying with AgInSe₂. Yet, the DFT calculations presented in fig. 5b do not seem to support this conclusion.

Response: We are sorry for this confusion. It is not obvious to distinguish the bandgap from the electronic density-of-states (Figure 5b). Actually, our calculation is consistent with the experimental result that the bandgap increases with introducing AgInSe₂ in PbSe matrix (Figure 5a). The theoretical bandgap for pure PbSe, Ag-doped PbSe, In-doped PbSe and Ag-In co-doped PbSe are compared in Figure S2d shown below. A possible reason for this phenomenon is given in our manuscript.

Fig. 5 Electronic band structure. **a** Electronic band structure of $Pb_{27}Se_{27}$ (black) and $Pb_{25}AgInSe_{27}$ (red). **b** Electronic density of states (DOS) near the Fermi level for $Pb_{27}Se_{27}$ (black), $Pb_{26}AgSe_{27}$ (green), $Pb_{26}InSe_{27}$ (blue) and $Pb_{25}AgInSe_{27}$ (red), respectively.

Supplementary Figure 2. Electronic band structures of (a) $\text{Pb}_{26}\text{AgSe}_{27}$ and (b) $\text{Pb}_{26}\text{InSe}_{27}$. (c) The energy offset (ΔE_{1-2}) between L and Σ valence band. (d) Theoretical bandgaps (E_g) for pristine, Ag-doped, In-doped and Ag-In co-doped PbSe.

Revision: The experimental bandgap is ~ 0.24 eV for the pristine PbSe, while the bandgap increases obviously with the incorporation of AgInSe₂ and a large bandgap ~ 0.33 eV is achieved for the PbSe - 2% AgInSe₂ sample (Figure 2d). The small bandgap of PbSe results from its unconventional chemical bonding mechanism (metavalent bonding). For a perfect half-filled p-band, the energy band structures resemble a metallic system. Yet, the bandgap opens due to a small Peierls distortion or charge transfer⁴¹. It is the charge transfer between Pb and Se that opens a small bandgap in PbSe given its perfect octahedral arrangements. DFT results show that the enlarged bandgap is mainly attributed to the incorporation of Ag. The electronegativity difference between Ag and Te (~ 0.62) is larger than that between Pb and Te (~ 0.22). Therefore, the substitution of Ag at Pb sites will strengthen the charge transfer between cation and anion, leading to an enlarged bandgap.

41. Yu, Y., Cagnoni, M., Cojocaru-Miré din, O. & Wuttig, M. Chalcogenide Thermoelectrics Empowered by an Unconventional Bonding Mechanism. *Adv. Funct. Mater.* **30**, 1904862 (2019).

Comment 3: Finally, in recent years the thermoelectric properties in lead chalcogenides have been discussed in terms of the underlying bonding mechanism, which must be related to the corresponding band structure [1,3]. A discussion of the fundamental bonding mechanism relevant here is missing. Such a discussion is important since it can help to predict and explain which materials and changes of bonding can improve the performance of a given thermoelectric material.

[1] Chemistry of Materials 32 (22), 9771-9779 (2020)

[3] Advanced Materials 32, 202005533 (2020)

Response: Thanks for your valuable suggestions. We have made discussions to understand the nature of electronic band structure and the bandgap behavior in chemical bonding perspective.

Revision:

Introduction section: Materials with disordered or complex crystal structure^{9,10}, giant anharmonicity^{11,12}, **metavalent bonding**¹³, and lone pair electrons¹⁴ often exhibit intrinsic low lattice thermal conductivity, which are promising candidates for thermoelectric applications.

Bandgap: The small bandgap of PbSe results from its unconventional chemical bonding mechanism (metavalent bonding). For a perfect half-filled p-band, the energy band structures resemble a metallic system. Yet, the bandgap opens due to a small Peierls distortion or charge transfer⁴¹. It is the charge transfer between Pb and Se that opens a small bandgap in PbSe given its perfect octahedral arrangements. DFT results show that the enlarged bandgap is mainly attributed to the incorporation of Ag. The electronegativity difference between Ag and Te (~ 0.62) is larger than that between Pb and Te (~ 0.22). Therefore, the substitution of Ag at Pb sites will strengthen the charge transfer between cation and anion, leading to an enlarged bandgap.

Electronic band structure: We calculated the orbital projected band structures to understand the nature of electronic band structure in chemical bonding perspective. The orbital projected band structures reveal that the interaction between Pb-p and Te-p orbitals dominate the band structure (Figure S3a, S3b), which is consistent with previous study⁴⁶. This is a typical feature of the metavalent bonding system^{13,45,46}. The Ag-d orbitals play an important role in modulating the third valence band Λ along Γ -L (Figure S3c). Our results indicate that Ag-In co-doping enable multiple valence band convergence, verifying that the cation-site doping is an effective way to modulate the valence band in PbSe. Similar effects can be expected in other materials, such as PbTe and GeTe, by employing the same chemical bonding mechanism as PbSe. A more detailed discussion is shown above when we answering the comment 1.

Temperature-dependent bandgap: Clearly, the bandgap increases with rising temperature, which is also verified experimentally (Figure 5d). As revealed by Brod et al.⁴⁴, there is sufficient interaction between Pb-p and Te-p (Se-p in our case) to

provide the molecular orbitals with the proper s-type symmetry to place the VBM at L point. The weak s-p hybridization is a small addition to this effect. The thermal expansion will lead to a reduction of orbital overlap between p-orbitals⁴¹ as well as a weakened s-p hybridization⁴⁸. As a result, the energy of VBM (L point) in the electronic band structure decreases⁴⁹, resulting in an enlarged bandgap (E_g).

13. Wuttig, M., Deringer, V. L., Gonze, X., Bichara, C. & Raty, J. Y. Incipient metals: functional materials with a unique bonding mechanism. *Adv. Mater.* **30**, e1803777 (2018).

41. Yu, Y., Cagnoni, M., Cojocaru-Miré din, O. & Wuttig, M. Chalcogenide Thermoelectrics Empowered by an Unconventional Bonding Mechanism. *Adv. Funct. Mater.* **30**, 1904862 (2019).

44. Brod, M. K., Toriyama, M. Y. & Snyder, G. J. Orbital chemistry that leads to high valley degeneracy in PbTe. *Chem. Mater.* **32**, 9771-9779 (2020).

45. Maier, S. *et al.* Discovering electron-transfer-driven changes in chemical bonding in lead chalcogenides (PbX, where X = Te, Se, S, O). *Adv. Mater.* **32**, e2005533 (2020).

46. Raty, J. Y. *et al.* A quantum-mechanical map for bonding and properties in solids. *Adv. Mater.* **31**, e1806280 (2019).

48. Zeier, W. G. *et al.* Thinking like a chemist: intuition in thermoelectric materials. *Angew. Chem. Int. Ed.* **55**, 6826-6841 (2016).

49. Cagnoni, M., Führen, D. & Wuttig, M. Thermoelectric performance of IV-VI compounds with octahedral-like coordination: a chemical-bonding perspective. *Adv. Mater.*, e1801787 (2018).

REVIEWERS' COMMENTS

Reviewer #1 (Remarks to the Author):

The authors have answered all the raised questions from these three reviewers. I am satisfied with the response. So, I suggest to accept this paper to publish in Nature Communications.

Reviewer #2 (Remarks to the Author):

The manuscript has been properly revised to address all the comments by the reviewers. Now it can be published as it is.

This work is a milestone for PbSe thermoelectrics, which possibly can outperform PbTe thermoelectrics in the near future.

Reviewer #3 (Remarks to the Author):

In the response to the questions and comments from the different reviewers, the authors have addressed all questions and concerns adequately. Hence, the manuscript is acceptable in its present form.